# Regional Business Environment, Agricultural Opening-Up and High-Quality Development: Dynamic Empirical Analysis from China's Agriculture

**Dezhen Wang [1,2], Buwajian Abula [1,*], Quan Lu [3,4], Yang Liu [5] and Yujiao Zhou [6,*]**

1    College of Economics and Management, Xinjiang Agricultural University, Urumqi 830052, China; dezhengumei@163.com
2    Business School, Yulin Normal University, Yulin 537000, China
3    College of Economics & Management, Huazhong Agricultural University, Wuhan 430077, China; luquan0122@163.com
4    College of Economics & Management, Tarim University, Alar 843300, China
5    Economics and Management School, Wuhan University, Wuhan 430072, China; 2019101050039@whu.edu.cn
6    School of Economics, Southwestern University of Finance and Economics, Chengdu 611130, China
*    Correspondence: buwajian@sina.cn (B.A.); zhouyujiao2015@163.com (Y.Z.)

**Abstract:** Agriculture is the foundation of every country's survival and development. This paper analyzes the interaction between the business environment, agricultural opening-up and high-quality agricultural economy from the perspective of China's provinces. According to the panel data of 31 provinces and municipalities in China from 2009 to 2019, the empirical analysis was carried out based on the panel vector autoregression (PVAR) model. The results show that there is a quantitative interaction between China's business environment, agricultural opening-up and high-quality agricultural development, and the long-term time effect is very obvious. Specifically, (1) the impact of the business environment and the high-quality development of agriculture on itself is decreasing year by year. The agriculture opening-up itself has a strong impact and has always maintained strong stability. (2) The high-quality development of the agricultural economy has little impact on the business environment and the opening of agriculture to the outside world, but it has a certain stable and continuous effect. It is particularly important that the agricultural opening-up has an increasingly strong effect on the business environment and the high-quality development of the agricultural economy as time progresses. Therefore, we must always adhere to the agriculture opening-up. The regional business environment and the agriculture opening-up can jointly promote the high-quality development of the agricultural economy. The research results can provide a basis and reference for the central and local governments to formulate relevant agricultural development policies and provide a reference for relevant agricultural economic entities and enterprises.

**Keywords:** regional business environment; agricultural opening-up; high-quality agricultural development; PVAR model

## 1. Introduction

Due to the profound and complex changes in the domestic and foreign economic situation caused by the new crown epidemic, China's economy has shifted from a stage of high-speed growth to a stage of high-quality development and is in a period of important strategic opportunities. At the same time, under the background that China's economy has entered a new development pattern with the domestic cycle as the main body and the domestic and international dual cycles driving each other, it means that the previous models, driving forces and methods of promoting economic development will change. The high-quality development of agriculture has become the key content for China's agricultural economy to shift from export to domestic demand. The regional business environment

is a necessary external environment for the high-quality development of agriculture, and the promotion of high-quality agricultural development is inseparable from the support of a high-quality business environment. At the same time, the agricultural opening-up is a necessary external driving force for the high-quality development of agriculture. The high-quality agricultural development can only operate well under the driving of both internal and external driving forces.

The new crown epidemic continues to spread and deepen its impact around the world [1,2]. New trade protections are on the rise. China faces a complex situation in which risks and challenges at home and abroad have increased significantly [3]. China has always insisted on optimizing the business environment as an important fulcrum for promoting high-quality economic development. The business environment is the internal core driving force and demand for promoting high-quality agricultural development. At the same time, with the continuous escalation of new trade protectionism and the continuous strengthening of food protection caused by the deepening of the new crown epidemic, the agricultural opening-up to the outside world has encountered unprecedented new dynamics. China's development is inseparable from the world, and the world's prosperity also needs China. China insists on opening to the outside world, especially in regard to agriculture. It helps to promote the high-quality development of the domestic economy, as well as brings many new opportunities for the development of the world economy, and provides more employment opportunities and commodities for enterprises and people around the world. These factors are bound to have a profound impact on the high-quality development of agriculture. Obviously, there must be some kind of internal relationship between the three factors. This article is the first time that these three factors are being placed into the same framework for a comprehensive analysis and discussion.

According to the natural attributes and characteristics of agricultural society, based on the panel data of Chinese provinces and cities from 2009 to 2020, and based on the panel vector autoregression (PVAR) model, this paper analyzes the interaction between business environment, agricultural opening-up and high-quality economy from the inter-provincial dimension. It can provide a basis and reference for the central and local governments to formulate relevant agricultural development policies.

## 2. Literature Review

High-quality economic development is an innovation-driven, comprehensive and sustainable growth mode. Therefore, the improvement of regional business environment and the agricultural opening-up are the connotations of high-quality economic development. Since the research on the high-quality development of agricultural economy has just started, there is not much research on the direct correlation between the three factors, and most of the literature discusses the business environment, agricultural opening-up and economic growth; economic growth is a very important aspect of high-quality development. Therefore, this chapter first sorts out the literature on high-quality agricultural development and then explores the business environment, agricultural opening-up and agricultural economic growth.

### 2.1. Literature Research on Regional Business Environment

According to Tomes research, many developed countries should also attach great importance to the construction of the business environment in the process of market construction [4]. Dobrin and Cioca concluded that the development strength of developing countries is relatively weak and the situation is different [5]. In the specific construction process, the local area should not intervene excessively in the construction and development of the business environment. The construction of the business environment has generally received strong support from government departments, and most developed countries have emphasized protecting the business environment. However, the construction of the business environment is not a short-term behavior, and all local parties should pay attention to long-term development and effects [6]. Kolasinski pointed out that the business

environment is the survival and development environment of enterprises, and it is the environment for investment and business in a region [7]. Haidar believed that Doing Business Report has four important purposes: First, it stimulates change by providing a benign benchmark for the economy. The second is to provide direction and goals for the reform and construction of the business environment. The third is to provide metrics for the flow of international development assistance. The fourth is to test the current theories related to policy supervision and management and economics, and promote the development and practical application of academic theories [8]. Xia et al. believed that the purpose of a good business environment is to speed up the introduction of enterprises in various provinces, and it is also the basis for enhancing the economic strength of the province [9]. Chen and Zhang believed that the purpose of optimizing the business environment is to provide a good external environment for the production and operation of enterprises and reduce the institutional transaction costs of the production and operation of enterprises [10].

*2.2. Literature Research on Opening-Up*

Tracing the origin of the theory of opening-up at home and abroad, Western economic research generally focuses on "whether opening-up can bring economic benefits to the country and the region". Balassa found that exports are beneficial to output growth [11]. Lawrence pointed to the need to be wary of over-opening in agricultural opening, that is, keeping agricultural cultivation and open borders to prevent embedding and new colonization [12]. Jones believed that the Heckscher–Ohlin trade theory promoted trade, and both sides can win related benefits [13]. Cakmac believes that agricultural development is greatly affected by structural changes and market opening [14]. Kydd pointed out that, only by participating in globalization and opening to the outside world can the small-scale peasant economy understand the world's advanced technologies and policies and promote its own long-term development [15]. Redding argued that the dynamics of comparative advantage will reduce the welfare level of one country [16]. Edsel found that relevant scholars have a common understanding when looking at the issue of "opening to the outside world will increase the income of the relevant countries"; that is, they all believed that the country is in contact with a larger market. The higher the degree of openness, the more economies of scale can be generated, so the development of foreign trade can improve a country's economic growth [17].

*2.3. Literature Research on High-Quality Agricultural Development*

First, research on the characteristics of high-quality agricultural development mainly explores its characteristics from the connotation and extension of high-quality agricultural development. The main features of high-quality agricultural development included green development leadership, supply quality and efficiency enhancement, large-scale production and diversified industrial integration [18]. Cheng et al. believed that the high-quality development of agriculture requires the benign interaction of technological progress, institutional changes and economic performance [19]. Ma and Hu divided the high-quality measures of agricultural economy into dimensions such as agricultural economic growth, agricultural economic structure, rural social development level and agricultural sustainability [20]. The view of Zhang and Liu is that the high-quality development of agriculture is the premise and foundation of rural revitalization, and it is the endogenous power support for rural revitalization [21].

Second is research on the status quo of high-quality agricultural development. Although my country's high-quality agricultural development is still in its infancy, many domestic and foreign experts and scholars have tried to explore its origins from different dimensions. Xia believed that the high-quality development of agriculture is not a small repair, but a major transformation of the development model and a major adjustment of the focus of work to promote a quality revolution in agriculture [22]. Chen pointed out that China's economy has shifted from a high-speed growth stage to a high-quality devel-

opment stage. The development of the agricultural economy has become an important engine to promote high-quality economic development [23]. Sun believed that the process of high-quality agricultural development still faces the contradiction between small-scale farmers' operation and large-scale production, the contradiction between high-cost input and increasing benefits, the contradiction between weak foundation and rapid development and the contradiction between pursuing high yield and environmental protection [24]. Hou et al. believed that the lack of a complete technology chain, the low level of technology maturity, the lack of technical connotation of brands and the lack of quality awareness of managers limit the high-quality development of agriculture [25].

Finally, we have the spatial econometric research on the high-quality development of agriculture. Although the research on spatial measurement of high-quality agricultural development has just started, it has been used by many researchers, due to its many excellent characteristics. Li and Zeng empirically analyzed the spatial spillover effect of China's agricultural infrastructure stock on agricultural economic growth by using the spatial data Durbin model [26]. The Moran spatial correlation test of Li and Chen shows that there are significant spatial spillover effects and diffusion effects in agricultural productivity growth and scientific research investment between provinces in China [27]. Although some experts and scholars have used spatial econometrics to study the high-quality development of agricultural economy, they rarely use PVAR to study the dynamic relationship between high-quality agricultural economy and other factors and variables.

*2.4. Research on the Relationship between Business Environment, Agricultural Opening-Up and High-Quality Agricultural Development*

There are few research studies in the literature on the relationship between business environment, agricultural opening-up and high-quality agricultural development. If the business environment is regarded as a part of total factor productivity, it is combined with agricultural product trade related to agricultural opening-up. There are the following studies. Yu and Liu studied the relationship between agricultural trade opening, technological progress and agricultural economic growth [28]. Wu studied the impact mechanism of international trade in agricultural products on economic growth and its impact on China's agricultural economic growth [29]. Yao believed that the contribution of environmentally friendly agricultural technology innovation activities to agricultural economic growth is growing steadily [30]. Liao and Zhang suggested to further improve the open agricultural policy, improve the linkage mechanism of international agricultural cooperation, further consolidate the strategic fulcrum of open agriculture and gradually form the entire industrial chain and value chain of open agriculture [31]. Ping and Wang used the co-integration theory and error correction model to empirically analyze the long-term impact and short-term impact of agricultural modernization and agricultural openness on farmers' income growth in Guizhou Province [32]. Feng believed that the increase of agricultural openness and the pulling effect of agricultural modernization on farmers' income will tend to be stable, and the contribution rate of agricultural openness to farmers' income increase is stronger than that of agricultural modernization [33].

In a word, the high-quality development of agriculture is accompanied by the pioneering proposal of high-quality economic development. Since there is no standardized and fixed paradigm, its research is also blooming. Scholars have explored from different angles and dimensions, but many studies lack systematicness and comprehensiveness, so it is difficult to describe the overall picture of high-quality agricultural development. This paper can better grasp the essence by studying the external environment (business environment) and internal driving force (agriculture opening up) of high-quality agricultural development. This paper is based on the above research to construct and analyze the system.

## 3. Index System Construction, Model Setting and Data Source

*3.1. The Construction of Index System*

3.1.1. Construction of Business Environment Index System

Starting from the connotation of the business environment, based on the research foundation of Liu and Xu [34] and the "Report on the Relative Progress of Marketization in China's Regions" by Fan et al., (2003), this paper introduces market allocation of resources, non-state-owned economic development [35]. Taking into account the scientific nature, systematicness and data availability of the index system, this paper constructs a regional business environment evaluation index system adapted to China's national conditions. The business environment indicator system is mainly set from six major dimensions: economic market size, public and private resources and opening-up, infrastructure, technology, human resources, finance, health and social security. It includes 14 secondary indicators and a total of 33 tertiary indicators, as shown in Table 1. In Table 1, the "Effect" column indicates the "+ (−)" means a positive (negative) index under the set measurement method, and the larger (smaller) it is, the better (worse) it is.

3.1.2. Construction of Agricultural Opening-Up Index System

The index of agricultural opening-up can construct a criterion layer from three aspects: the output of main agricultural products per capita, agricultural import and export, and the quality of living standards. Among them, the quality of living is composed of the per capita output of six essential agricultural products. Generally, if a country or region has insufficient local production of these agricultural products, it must be imported from foreign countries, which can better reflect the fact that agriculture is open to the outside world. The factor layer further refines the classification indicators. The measurement indicators of the main agricultural product output can be composed of per capita agricultural products such as grain, milk, aquatic products, edible oil, and pig, beef and mutton output. These indicators are closely related to people's daily life. Four indicators are used to measure the export value of agricultural products for agricultural import and export, the import value of agricultural products, the export value of agricultural factors input, and the import volume of agricultural factors input. The demand for imported agricultural products is mainly determined by the income and consumption of a country and region. Therefore, the per capita disposable income of urban residents and the per capita consumption expenditure of rural residents are used to measure the income, consumption and living standards. The measure indicator layer is specific to each indicator. (See Table 2 for details, where the weights are those calculated by the entropy method later).

3.1.3. Construction of Agricultural High-Quality Development Index System

Considering the availability and continuity of data, this paper constructs a measurement system for the level of agricultural high-quality development. First, we measure the pros and cons by comparing the construction of high-quality agricultural development at home and abroad and select the indicators that best represent the essence of high-quality agricultural development to construct the criterion layer. Second, the high-quality results in the four directions of high-quality agricultural development are extended to high-quality elements, and the dimensions of high-quality elements have different perspectives to reflect the scientific connotation of high-quality agricultural development. Finally, each specific measurement index is refined according to 11 high-quality factor dimensions. According to the measurement indicators, the effect of high-quality agricultural development is divided, and the 25 specific measurement indicators are divided into 21 positive indicators and 4 negative indicators; see Table 3 for details. The combined 10-year average weight is placed after each level indicator.

**Table 1.** Evaluation index system and weight composition of China's provincial and municipal business environment.

| First-Level Indicators | Weights (%) | Secondary Indicators | Three-Level Indicators | $X_{ij}$ | Weights (%) | Effect |
|---|---|---|---|---|---|---|
| Economic environment | 0.0975 | Economic strength | GDP per capita (yuan) | $X_1$ | 0.0227 | + |
| | | | GDP growth rate (%) | $X_2$ | 0.0083 | + |
| | | | Fixed asset investment per capita (yuan) | $X_3$ | 0.0144 | + |
| | | Economic scale | Per capita consumption expenditure of urban residents (yuan) | $X_4$ | 0.0298 | + |
| | | | Total retail sales of social consumer goods per capita (yuan) | $X_5$ | 0.0229 | + |
| | | | Fiscal expenditure/GDP (%) | $X_6$ | 0.0354 | + |
| Market environment | 0.1857 | Market resource allocation | Proportion of private industrial enterprises in the profits of industrial enterprises above designated size (%) | $X_7$ | 0.0135 | + |
| | | | Proportion of non-state-owned enterprises in fixed asset investment (%) | $X_8$ | 0.0042 | + |
| | | Non-state economy | Proportion of the main business income of private industrial enterprises in the main business income of industrial enterprises above designated size (%) | $X_9$ | 0.0128 | + |
| | | Opening-up | Total investment by foreign-invested enterprises (US $100 million) | $X_{10}$ | 0.0513 | + |
| | | | Total import and export goods (billion US dollars) (by domestic destination) | $X_{11}$ | 0.0677 | + |
| Infrastructure | 0.2422 | Transportation | Density of graded highways (high-speed, first-class, second-class) (km/10,000 km$^2$) | $X_{12}$ | 0.0163 | + |
| | | | Railway density (km/10,000 km$^2$) | $X_{13}$ | 0.0193 | + |
| | | | Employment of postal personnel per 10,000 people (person/10,000) | $X_{14}$ | 0.0333 | + |
| | | | Cargo turnover (100 million ton-kilometers/10,000 km$^2$) | $X_{15}$ | 0.1430 | + |
| | | Living facilities | Water supply per capita (m$^3$/person) | $X_{16}$ | 0.0335 | + |
| Technology, human capital and financial environment | 0.2265 | Technology development | Invention, utility model and design patent applications (items) | $X_{17}$ | 0.0467 | + |
| | | | Technology market turnover (100 million yuan) | $X_{18}$ | 0.0718 | + |
| | | Labor | Average number of undergraduate students per 10,000 population (person) | $X_{19}$ | 0.0106 | + |
| | | | Average wages of urban employees (yuan) (arithmetic average of urban employees in non-private and private units) | $X_{20}$ | 0.0257 | + |
| | | Financial environment | Employment in the financial industry (ten thousand people) | $X_{21}$ | 0.0170 | + |
| | | | The ratio of gross wages of financial workers to GDP | $X_{22}$ | 0.0449 | + |
| | | tax burden | Proportion of corporate income tax to tax revenue (%) | $X_{23}$ | 0.0061 | − |
| Government affairs judicial environment | 0.1547 | Government environment | Leasing and business services (person/ten thousand people) | $X_{24}$ | 0.0771 | + |
| | | | Water conservancy, environment and public facilities management (person/ten thousand people) | $X_{25}$ | 0.0161 | + |
| | | | Culture, sports and entertainment (person/ten thousand people) | $X_{26}$ | 0.0486 | + |
| | | Judicial environment | Average direct property loss per person in traffic accident (yuan/person) | $X_{27}$ | 0.0063 | − |
| | | | Proportion of confiscation revenue to general budget (public finance) revenue (%) | $X_{28}$ | 0.0077 | − |
| Health and social security | 0.0933 | Health and social security | Number of health technicians per 1000 population | $X_{29}$ | 0.0169 | + |
| | | | Beds in medical and health institutions per 1000 population (number, average in urban and rural areas) | $X_{30}$ | 0.0142 | + |
| | | | Proportion of participating in urban basic endowment insurance (%) | $X_{31}$ | 0.0164 | + |
| | | | Proportion of people participating in unemployment insurance (%) | $X_{32}$ | 0.0287 | + |
| | | | Proportion of urban basic medical insurance participants (%) | $X_{33}$ | 0.0172 | + |

Note: "+ (−)" in the column of "Effect" indicates that the measurement index is a positive (negative) index under the set measurement method, and the larger (smaller) is, the better (bad) is.

**Table 2.** Measurement system of China's agriculture opening-up.

| Criterion Layer | Element Layer | Element Layer Weight (%) | Measurement Index | Index Measurement Unit | $X_{ij}$ | Effect | Measurement Index Weight (%) |
|---|---|---|---|---|---|---|---|
| Per capita output of main agricultural products | Basic living security | 0.0302 | Other grain production per capita | Kg | $X_1$ | − | 0.0155 |
| | | | Per capita cereal production | Kg | $X_2$ | − | 0.0147 |
| | Quality-of-life improvement | 0.0486 | Per capita edible oil production | Kg | $X_3$ | − | 0.0209 |
| | | | Per capita production of pork, beef and mutton | Kg | $X_4$ | − | 0.0277 |
| | | 0.0260 | Per capita output of aquatic products | Kg | $X_5$ | − | 0.0155 |
| | | | Milk production per capita | Kg | $X_6$ | − | 0.0105 |
| Agricultural import and export | Import and export of agricultural products | 0.4080 | Export volume of agricultural products | $ | $X_7$ | + | 0.2129 |
| | | | Imports of agricultural products | $ | $X_8$ | + | 0.1951 |
| | Import and export of agricultural elements | 0.3377 | Agricultural factor input and export | $ | $X_9$ | + | 0.1404 |
| | | | Agricultural factor input imports | $ | $X_{10}$ | + | 0.1973 |
| Quality of living standard | Income and consumption standard | 0.1495 | Per capita disposable income of urban residents | Yuan | $X_{11}$ | + | 0.1001 |
| | | | Per capita consumption expenditure of rural residents | Yuan | $X_{12}$ | + | 0.0494 |

Note: "+ (−)" in the column of "Effect" indicates that the measurement index is a positive (negative) index under the set measurement method, and the larger (smaller) is, the better (bad) is.

**Table 3.** Agricultural high-quality development level measurement system.

| High Quality Result Layer Weights (%) | High Quality Feature Layer Weight (%) | Measure Indicator Layer Weight (%) | Indicator Measurement Layer | $N_{ij}$ | Effect |
|---|---|---|---|---|---|
| High quality, high yield and high efficiency (0.2118) | Food production capacity (0.0609 $X_1$) | Food production population ratio (0.0365) | Total grain output/regional population (tons/person) | $N_1$ | + |
| | | Grain yield per unit of arable land (0.0244) | Regional total cereal production/arable land (10,000 tons/1000 hectares) | $N_2$ | + |
| | Production capacity guarantee (0.0750 $X_2$) | Agricultural machinery ownership ratio (0.0352) | Total power of agricultural machinery (10,000 kilowatts)/arable land area (1000 hectares) | $N_3$ | + |
| | | Ratio of cultivated land irrigated area (0.0397) | Irrigated area of arable land/area of arable land (percentage) | $N_4$ | + |
| | | Growth rate of added value of primary industry (0.0079) | (The added value of the primary industry this year/the added value of the primary industry in the previous year-1) × 100% | $N_5$ | + |
| | Production efficiency (0.0759 $X_3$) | Labor productivity (0.0234) | Gross output value of agriculture, forestry, animal husbandry and fishery/rural population (100 million yuan/10,000 people) | $N_6$ | + |
| | | Productivity of arable land (0.0446) | Gross agricultural output value/arable land area (100 million yuan/1000 hectares) | $N_7$ | + |
| High-efficiency agriculture (0.4096) | Production Benefit (0.1128 $X_4$) | Urban–rural income ratio (0.0160) | Per capita disposable income of urban residents/per capita disposable income of rural residents | $N_8$ | - |
| | | Average wage of agricultural employees in urban private units (0.0305) | Average wages of persons employed in agriculture, forestry, animal husbandry and fishery in urban private units (unit: yuan) | $N_9$ | + |
| | | Per capita disposable income of rural residents (0.0373) | Unit: yuan | $N_{10}$ | + |
| | | Per capita consumption expenditure of rural residents (0.0290) | Unit: yuan | $N_{11}$ | + |

**Table 3.** *Cont.*

| High Quality Result Layer Weights (%) | High Quality Feature Layer Weight (%) | Measure Indicator Layer Weight (%) | Indicator Measurement Layer | $N_{ij}$ | Effect |
|---|---|---|---|---|---|
| | Put investment (0.0990 $X_5$) | Agricultural input-output ratio (0.0673) | Added value of primary industry/total investment in agriculture, forestry, animal husbandry and fishery | $N_{12}$ | + |
| | | Agriculture, forestry and water budget expenditure ratio (0.0174) | Agriculture, forestry and water budget expenditure/local general public budget expenditure | $N_{13}$ | + |
| | | Growth rate of fixed asset investment in agriculture, forestry, animal husbandry and fishery (excluding farmers) (0.0144) | (Current year value/last year value-1) × 100 | $N_{14}$ | + |
| | Quality of life (0.0112 $X_6$) | The average number of washing machines owned by rural residents per 100 households at the end of the year (0.0112) | Representatives of major durable consumer goods (Taiwan) | $N_{15}$ | + |
| | Competitiveness (0.1866 $X_7$) | Agricultural product export ratio (0.1866) | Exports of agricultural products/agricultural GDP (USD/100 million yuan) | $N_{16}$ | + |
| Modern agricultural management system (0.3218) | Optimization of modern agricultural organization system (0.1514 $X_8$) | Proportion of rural population (0.0137) | Rural population/total population (1-the proportion of urban population)% | $N_{17}$ | − |
| | | Corporate ratio of agriculture, forestry, animal husbandry and fishery (0.0304) | Number of legal persons in agriculture, forestry, animal husbandry and fishery/number of legal entities in all industries | $N_{18}$ | + |
| | | Ratio of employed persons in non-private units in agriculture, forestry, animal husbandry and fishing cities and towns (0.1073) | Number of employed persons/total employed persons in non-private units in agriculture, forestry, animal husbandry and fishery | $N_{19}$ | + |
| | Industrial integration level (0.1704 $X_9$) | Employment ratio of rural private enterprises (0.0267) | Employment in rural private enterprises/total employment in private enterprises (by region) | $N_{20}$ | + |
| | | Private enterprise employment rate of rural population (0.1437) | Rural private enterprise employment/rural population | $N_{21}$ | + |
| Green sustainable development (0.0569) | Resource consumption efficiency (0.0239 $X_{10}$) | Agricultural fertilizer application rate (0.0158) | Agricultural chemical fertilizer application amount (10,000 tons)/arable land area (1000 hectares) | $N_{22}$ | − |
| | | Water consumption per ten thousand yuan of agricultural GDP (0.0080) | Agricultural water consumption/agricultural added value (cubic meter/yuan) | $N_{23}$ | − |
| | Prevention and sanitation (0.0330 $X_{11}$) | Forest pest control situation Control rate (0.0131) | (Total number of pests, pests and harmful plants) (%) | $N_{24}$ | + |
| | | Rural medical level (0.0199) | Rural doctors and health workers per 10,000 people (rural population, unit: number) | $N_{25}$ | + |

### 3.2. Calculation Method and Model Setting

3.2.1. Calculation Method

Drawing on the research methods of references [36–40], this paper adopts the entropy method is used to measure the comprehensive evaluation index value of business environment, agricultural opening-up and agricultural high-quality development. Specific steps are as follows:

(1) Standardized processing of data.

Due to the differences in dimensions and units among the indicators, the data should be standardized before determining the weights. The processing formula is as follows:

$$r_{ij} = \left[ \frac{x_{ij} - \min x_{ij}}{\max x_{ij} - \min x_{ij}} \right] * 0.9 + 0.1 \text{ or } r_{ij} = \left[ \frac{\max x_{ij} - x_{ij}}{\max x_{ij} - \min x_{ij}} \right] * 0.9 + 0.1 \quad (1)$$

Among them, $r_{ij}$ is the standard index value; the range is [0.1, 1]; $x_{ij}$ refers to the actual value of the $j$th index in the $i$ province; and min $x$ and max $x$ refer to the minimum and maximum values of the $j$ index, respectively. Positive indicators are processed by using the former formula, while negative indicators are processed by using the latter formula.

(2)　Calculate the proportion of the sample value of the province $i$ under the $j$ index.

$$p_{ij} = r_{ij} / \sum_{i=1}^{m} r_{ij} \ (i = 1, \ 2, \ \ldots, \ m; \ j = 1, \ 2, \ \ldots, \ n) \tag{2}$$

(3)　Calculation of the entropy value of the $j$ index.

$$e_j = \frac{1}{\ln m} \sum p_{ij} \ln p_{ij} \ (i = 1, \ 2, \ \ldots, \ m; \ j = 1, \ 2, \ \ldots, \ n) \ (0 \le e_{ij} \le 1) \tag{3}$$

(4)　Calculation of the coefficient of difference of the $j$ index.

The degree of difference changes inversely with the entropy value. The smaller the entropy value, the greater the degree of difference between the indicators, the greater the amount of information reflected and the greater the impact and role of the indicators on the evaluation object.

$$f_j = 1 - e_j \ (j = 1, 2, \ldots, n) \tag{4}$$

(5)　Determination of the weight of the $j$ indicator.

$$w_j = f_j / \sum_{j=1}^{n} f_j \ (j = 1, \ 2, \ \ldots, \ n) \tag{5}$$

After the corresponding weights of the indicators are determined, the comprehensive evaluation value of the indicator system is calculated according to Equation (6).

$$\mathrm{S} = \sum p_{ij} w_{ij} \ (i = 1, \ 2, \ \ldots, \ m; \ j = 1, \ 2, \ \ldots, \ n) \tag{6}$$

3.2.2. Model Setting

In this paper, the panel vector autoregression (PVAR) model can use dynamic panel data GMM estimation and impulse response dynamic analysis. Then variance decomposition prediction analysis is used to measure the future spatial relationship and time effects of China's high-quality agricultural economic development, business environment and agricultural opening-up. Therefore, this paper selects a dynamic panel model with fixed effects, and all variables can be regarded as endogenous variables. In other words, the panel vector autoregression model not only has the advantages of the panel data estimation method, but also has the advantages of the vector autoregressive model, and can also better capture the influence of each sample individual on the model parameters; thus, the following model is established:

$$Y_{it} = \gamma_0 + \sum_{j=1}^{P} \gamma_j Y_{it-j} + \alpha_i + \beta_t + \varepsilon_{it} \tag{7}$$

Among them, i represents different provinces, regions and cities; t represents the year; $Y_{it}$ includes three column vectors, namely the high-quality development of the agricultural economy, business environment and agricultural opening-up; $\gamma_0$ represents the intercept term vector; P represents the lag order; $\gamma_j$ represents the parameter matrix of lag j order; $\alpha_i$ represents the individual fixed-effect variable; $\beta_t$ represents the time-effect variable; and $\varepsilon_{it}$ represents the random disturbance term.

*3.3. Data Sources and Indicator Calculation Results*

3.3.1. Data Sources

This paper selects the panel data of 31 provinces and municipalities in China from 2009 to 2018 as the research sample. The data come from the 2010–2019 "China Statistical Yearbook", China Rural Statistical Yearbook, China Agricultural Statistical Yearbook,

60 Years of New China Statistical Data Compilation, Provincial and Municipal Statistical Yearbooks, Customs Statistical Data and relevant statistical data publicly released by the Information Network of the Development Research Center of the State Council. For the problem of missing data, this paper uses the interpolation method to supplement it.

3.3.2. Indicator Calculation Results

The statistics of the indicator calculation results are given in the form of Table 4.

**Table 4.** Scores and rankings of high-quality agricultural development, business environment and agricultural opening-up in various provinces.

| Item Region | Comprehensive Score of Agricultural High-Quality Development | Comprehensive Ranking of Agricultural High-Quality Development | Business Environment Comprehensive Level Score | Comprehensive Ranking of Business Environment | Comprehensive Score of Agricultural Opening-Up | Ranking of Comprehensive Level of Agricultural Opening-Up |
|---|---|---|---|---|---|---|
| Beijing | 0.4682 | 2 | 0.9913 | 1 | 0.4758 | 6 |
| Tianjin | 0.2621 | 10 | 0.3740 | 6 | 0.3087 | 8 |
| Hebei | 0.2228 | 11 | 0.0943 | 21 | 0.1952 | 11 |
| Shanxi | 0.0946 | 24 | 0.0685 | 25 | 0.0747 | 16 |
| Neimenggu | 0.2173 | 14 | 0.1293 | 15 | 0.0527 | 26 |
| Liaoning | 0.1977 | 16 | 0.2419 | 8 | 0.2496 | 9 |
| Jilin | 0.1668 | 20 | 0.1012 | 18 | 0.0625 | 23 |
| Heilongjiang | 0.3619 | 5 | 0.0892 | 23 | 0.0693 | 18 |
| Shanghai | 1.0000 | 1 | 0.9310 | 2 | 0.6297 | 5 |
| Jiangsu | 0.3923 | 4 | 0.4872 | 4 | 0.8935 | 2 |
| Zhejiang | 0.4331 | 3 | 0.3979 | 5 | 0.7283 | 4 |
| Anhui | 0.1853 | 18 | 0.0980 | 20 | 0.1173 | 14 |
| Fujian | 0.2958 | 7 | 0.1904 | 10 | 0.3701 | 7 |
| Jiangxi | 0.2120 | 15 | 0.0591 | 26 | 0.0612 | 24 |
| Shandong | 0.3172 | 6 | 0.2531 | 7 | 0.7527 | 3 |
| Henan | 0.1868 | 17 | 0.0922 | 22 | 0.0672 | 19 |
| Hubei | 0.1679 | 19 | 0.1499 | 11 | 0.1251 | 13 |
| Hunan | 0.2175 | 13 | 0.1154 | 17 | 0.0730 | 17 |
| Guangdong | 0.2224 | 12 | 0.5202 | 3 | 0.8941 | 1 |
| Guangxi | 0.0956 | 23 | 0.0459 | 28 | 0.2243 | 10 |
| Hainan | 0.2870 | 8 | 0.0844 | 24 | 0.0385 | 27 |
| Chongqing | 0.0972 | 22 | 0.1974 | 9 | 0.0791 | 15 |
| Sichuan | 0.1096 | 21 | 0.1190 | 16 | 0.0579 | 25 |
| Guizhou | 0.0611 | 28 | 0.0249 | 29 | 0.0668 | 21 |
| Yunnan | 0.0634 | 27 | 0.0135 | 30 | 0.1398 | 12 |
| Xizang | 0.0553 | 29 | 0.1003 | 19 | 0.0039 | 31 |
| Shaanxi | 0.0486 | 30 | 0.1328 | 13 | 0.0670 | 20 |
| Gansu | 0.0190 | 31 | 0.0042 | 31 | 0.0235 | 29 |
| Qinghai | 0.0822 | 26 | 0.0548 | 27 | 0.0095 | 30 |
| Ningxia | 0.0890 | 25 | 0.1312 | 14 | 0.0269 | 28 |
| Xinjiang | 0.2801 | 9 | 0.1406 | 12 | 0.0638 | 22 |

## 4. Results and Discussion

### 4.1. Panel Unit Root Stability Test and Optimal Order

This paper uses the PVAR model to analyze the high-quality development of agricultural economy (NYGZ), business environment (YS) and agricultural opening to the outside world (KF) (the first-order difference of their respective 2009–2020 comprehensive ranking score data are represented by AH, BE and AO, respectively, compared with the second-order difference processing results). Before conducting the research on the interaction effect, this study determined that the optimal lag order of the first-order difference of the PVAR model is 2, and the optimal lag order of the second-order difference is, 1 according to the AIC, BIC and HQIC criteria. See Table 5 for details.

**Table 5.** Selection of optimal lag order.

| | First-Order Difference of Data | | | Second Order Difference of Data | | |
|---|---|---|---|---|---|---|
| Lag | AIC | BIC | HQIC | AIC | BIC | HQIC |
| 1 | −13.8618 | −12.4167 | −13.2801 | −12.3963 * | −10.8076 * | −11.7545 * |
| 2 | −14.3356 * | −12.6067 * | −13.6372 * | −11.8734 | −9.9484 | −11.0933 |
| 3 | −14.1364 | −12.0552 | −13.2930 | −11.2496 | −8.8934 | −10.2926 |
| 4 | −13.8206 | −11.2877 | −12.7918 | −10.2228 | −7.2888 | −9.0310 |
| 5 | −13.2713 | −10.1326 | −11.9963 | −9.6410 | −5.8829 | −8.1236 |

Notes: Standard errors are in parentheses; * $p < 0.05$.

Before estimating the PVAR model, we should first determine whether each variable is a stationary series. Therefore, this paper uses HT, Fisher, Breitung and Hadri tests, four common panel data unit root test methods, to perform a unit root test on the above variables, in turn. The test results after the first-order difference and the first-order lag processing of each variable reject the null hypothesis of "there is a unit root" at the 1% significance level, that is, each variable after the difference is a stationary sequence. As shown in the left of Figure 1, it can be seen that the modulus of the eigenvalues of the adjoint matrix are all less than 1. Each variable after the first-order difference and second-order lag processing cannot reject the null hypothesis of "there is a unit root" at the 1% significance level; that is, each variable after the difference is a non-stationary sequence. As can be seen in Figure 1, on the right, the modulus of the eigenvalues of the adjoint matrix is greater than 1.

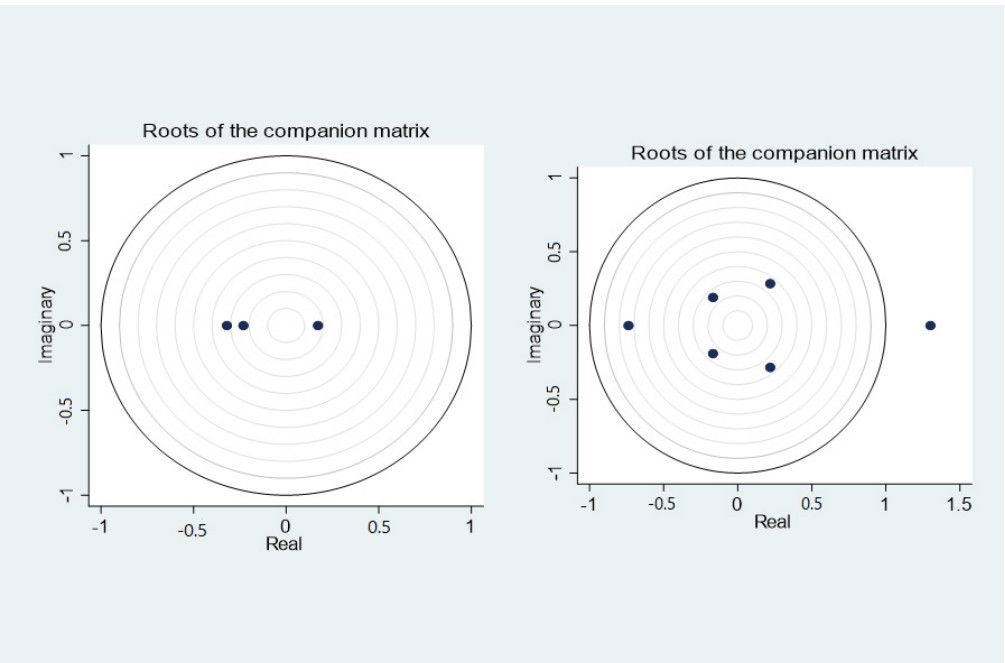

**Figure 1.** Stationarity test of the PVAR model (left is first-order difference first-order lag, and right is first-order difference second-order lag).

The second-order difference data are also tested. After the second-order difference and the first-order lag processing of the test results, the null hypothesis of "there is a unit root" is rejected at the 1% significance level; that is, the variables after the difference are stationary series. As can be seen from the left of Figure 2, each modulus of the eigenvalues of the adjoint matrix is less than 1. Each variable after the second-order difference and second-order lag processing can also reject the null hypothesis of "there is a unit root" at the 1% significance level; that is, each variable after the difference is also a stationary sequence. As can be seen from the right of Figure 2, each modulus of the eigenvalues of the

adjoint matrix is less than 1. Compared with Figure 1, Figure 2 more intuitively shows that some characteristic roots are located within the unit circle to satisfy the stability condition. In summary, the data processed by the second-order difference and the first-order lag are more suitable for constructing a PVAR model.

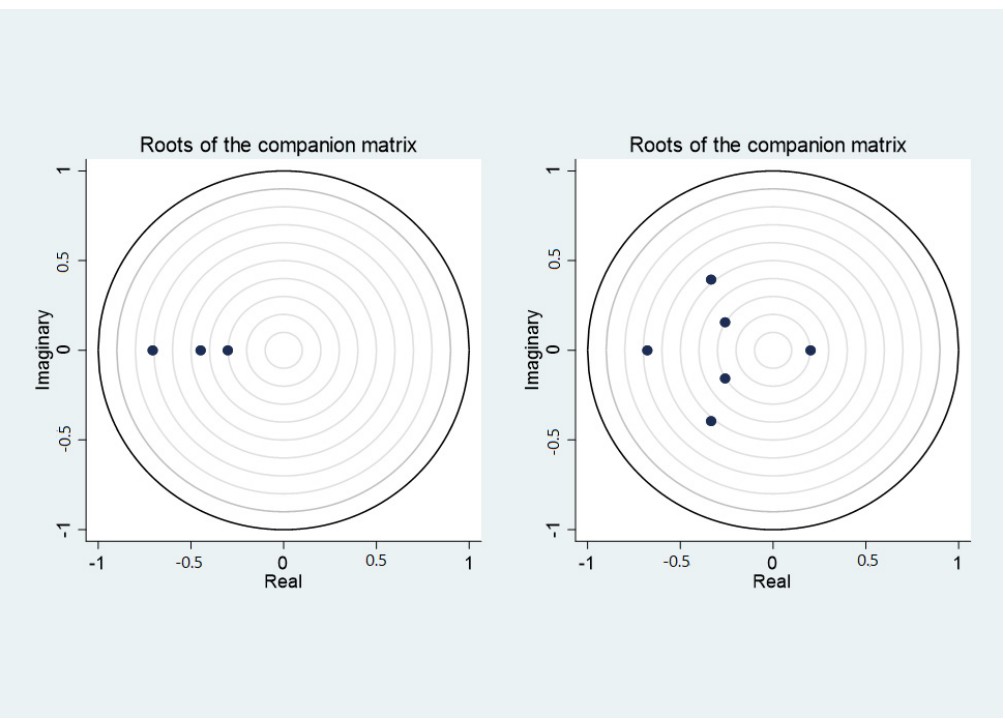

**Figure 2.** Stationarity test of the PVAR model (the left is the second-order difference first-order lag, and the right is the second-order difference second-order lag).

### 4.2. Granger Causality Test

In Table 6, the first row on the left gives the Wald test of whether the coefficients of 1 lag period of the business environment (YS) in the PVAR equation are jointly 0. The *p*-value is 0.006, thus rejecting the null hypothesis that the coefficients of the two lags of the business environment (YS) are jointly 0. Thus, it can be considered that the business environment (YS) is the Granger of Agricultural Quality Development (NYGZ). The second row on the left gives the Wald test of whether the coefficients of the two lags of agricultural opening-up (KF) in the PVAR equation are jointly 0. According to the corresponding *p*-value of 0.759 in Table 6, it cannot be considered that the agricultural opening-up (KF) is the Granger of the agricultural high-quality development (NYGZ). The third row on the left gives the Wald test of whether the coefficients of the two lag periods of all other endogenous variables in the PVAR equation are jointly 0. The *p*-value is 0.005; thus, the business environment (YS) and agricultural opening-up (KF) together can be considered the Granger of the high-quality development of the agricultural economy (NYGZ). We can make a similar interpretation for the other equations in Table 6; see Table 6 for details.

### 4.3. Dynamic Panel Data GMM Estimation

The PVAR model uses the lag term of the endogenous variable as an instrumental variable and uses the systematic GMM method to eliminate the endogeneity in the model. The time effect and individual effect in the model are removed by the cross-sectional mean difference method and the forward mean difference method (Helmert). Among them, the sequences of NYGZ, YS and KF (AH, BE and AO) after Helmert transformation to eliminate individual effects are h_NYGZ, h_YS and h_KF (h_AH2, h_BE2, h_AO2), respectively. L represents the variable with a lag of one period. See Table 7 for details.

**Table 6.** PVAR–Granger causality Wald test.

| Equation | Excluded | chi2 | df | Prob > chi2 | Equation | Excluded | chi2 | df | Prob > chi2 |
|----------|----------|------|----|-------------|----------|----------|------|----|-------------|
| NYGZ | YS | 7.632 | 1 | 0.006 | AH2 | BE2 | 0.110 | 1 | 0.740 |
| NYGZ | KF | 0.094 | 1 | 0.759 | AH2 | AO2 | 0.000 | 1 | 0.986 |
| NYGZ | ALL | 10.413 | 2 | 0.005 | AH2 | ALL | 6.193 | 2 | 0.908 |
| YS | NYGZ | 0.629 | 1 | 0.428 | BE2 | AH2 | 4.895 | 1 | 0.027 |
| YS | KF | 0.142 | 1 | 0.706 | BE2 | AO2 | 1.037 | 1 | 0.308 |
| YS | ALL | 0.655 | 2 | 0.721 | BE2 | ALL | 5.337 | 2 | 0.069 |
| KF | NYGZ | 0.606 | 1 | 0.436 | AO2 | AH2 | 0.835 | 1 | 0.361 |
| KF | YS | 0.030 | 1 | 0.863 | AO2 | BE2 | 0.141 | 1 | 0.708 |
| KF | ALL | 1.821 | 2 | 0.402 | AO2 | ALL | 1.011 | 2 | 0.603 |

**Table 7.** GMM estimation results of PVAR model (lag 1 period).

| VAR (341) | h_NYGZ | h_YS | h_KF | VAR (217) | h_AH2 | h_BE2 | h_AO2 |
|-----------|--------|------|------|-----------|-------|-------|-------|
| L.h_NYGZ | 0.4197 (1.1923) | 0.0978 (0.4027) | 0.1574 (1.8927) | L.h_AH2 | −0.5041 *** (0.0846) | 0.0204 (0.0403) | −0.1390 (0.1152) |
| L.h_YS | −1.8264 (4.4165) | 1.1772 (1.4800) | 0.6750 (6.9853) | L.h_BE2 | 0.0623 (0.1848) | −0.3091 *** (0.0710) | −0.1257 (0.2208) |
| L.h_KF | 0.0365 (0.0872) | 0.0039 (0.0264) | 0.2983 (0.1615) | L.h_AO2 | −0.0202 (0.0562) | −0.0192 (0.0141) | −0.3958 *** (0.0717) |

Notes: Standard errors are in parentheses; *** $p < 0.001$.

From the GMM estimation results of the PVAR model in Table 7, it can be seen that removing the individual effects h_AH2, h_BE2 and h_AO2 from the model is more significant than removing the time effects h_NYGZ, h_YS and h_KF from the model. Among them, the interaction of h_AH2, h_BE2 and h_AO2 (current period and one lag period) is very significant, showing that h_AH2, h_BE2 and h_AO2 have significant effects on themselves. At the same time, the second-order difference data on the right side of Table 7 are more significant than the original data on the left side of Table 7. It shows that the h_AH2, h_BE2 and h_AO2 data without the individual effects in the model are more significant than the original data without removal. It also shows that NYGZ (High Quality Agricultural Development), YS (business environment) and KF (agricultural opening to the outside world) have strong correlations over the years.

By comparing Tables 7 and 8, it can be seen that the first lag (0.7133 **) of the h_KF interaction is significant, but the second lag (0.2983 *) is generally significant. The first lag (−0.6065 ***) of the h_AH interaction was very significant, but its second lag (−0.1502) was not. The first lag (−0.3067 ***) of the h_BE interaction was very significant, but its second lag (−0.0730) was not. All three indicate that their effect on itself is attenuated with a time lag. At the same time, by comparing the data on the left and right sides of Table 8, it can be seen that the difference (excluding the individual effect) is more significant than that without the difference. It can also be seen from Tables 7 and 8 that the three have significant negative values for the lag period itself and other variables after removing the individual effect. It suggests that they have an inhibitory effect on subsequent development.

### 4.4. Impulse Response Dynamic Analysis

The GMM estimation results show the static relationship among NYGZ, YS and KF (AH, BE and AO). In order to obtain a more accurate dynamic relationship, the dynamic analysis of impulse response of NYGZ, YS and KF (AH2, BE2, AO2) is carried out in this paper. We used the Monte Carlo algorithm to perform 600 simulations to estimate the impulse response function plots with 95% confidence intervals for each variable, with a lag of 10 periods (years). As shown in Figure 3 (the middle line in the figure is the estimated value of the IRF point, and the upper and lower lines represent the upper and lower boundary lines of the 95% confidence interval). The horizontal axis represents the number of lag periods. This paper mainly studies the interactive relationship between

high-quality agricultural development, business environment and agricultural opening to the outside world. Therefore, we mainly explore the response of agricultural high-quality development (NYGZ) to the impulse shocks from the business environment (YS) and agricultural opening-up (KF), as shown in row one of Figure 3, and the responses of YS and KF to the shock from the NYGZ pulse, as shown in column one of Figure 3. Among them, IRF is the English abbreviation of impulse response function.

**Table 8.** GMM estimation results of PVAR model (lag 2).

| VAR (341) | h_NYGZ | h_YS | h_KF | VAR (279) | h_AH | h_BE | h_AO |
|---|---|---|---|---|---|---|---|
| L.h_NYGZ | 0.4197 (1.1923) | 0.0978 (0.4027) | 0.1574 (1.8927) | L.h_AH | −0.6065 *** (0.1419) | 0.0342 (0.0441) | −0.1851 (0.1211) |
| L.h_YS | −1.8264 (4.4165) | 1.1772 (1.4800) | 0.6750 (6.9853) | L.h_BE | 0.2816 (0.3037) | −0.3067 *** (0.0786) | −0.1252 (0.2548) |
| L.h_KF | −0.1167 (0.2591) | 0.0405 (0.0693) | 0.7133 ** (0.3296) | L.h_AO | −0.0808 (0.1819) | −0.0490 ** (0.0209) | −0.5240 *** (0.1147) |
| L2.h_NYGZ | 0.1566 (0.2342) | −0.0497 (0.0694) | −0.0650 (0.3071) | L2.h_AH | −0.1502 (0.1281) | −0.0041 (0.0412) | 0.0316 (0.1355) |
| L2.h_YS | −0.0479 (0.3798) | −0.1036 (0.1489) | 0.0296 (0.5668) | L2.h_BE | 0.3233 (0.2656) | −0.0730 (0.0861) | −0.1808 (0.2888) |
| L2.h_KF | 0.0365 (0.0872) | 0.0039 (0.0264) | 0.2983 * (0.1615) | L2.h_AO | −0.05466 (0.0826) | −0.0461 ** (0.0158) | −0.2658 *** (0.0583 |

Notes: Standard errors are in parentheses; * $p < 0.05$, ** $p < 0.01$ and *** $p < 0.001$.

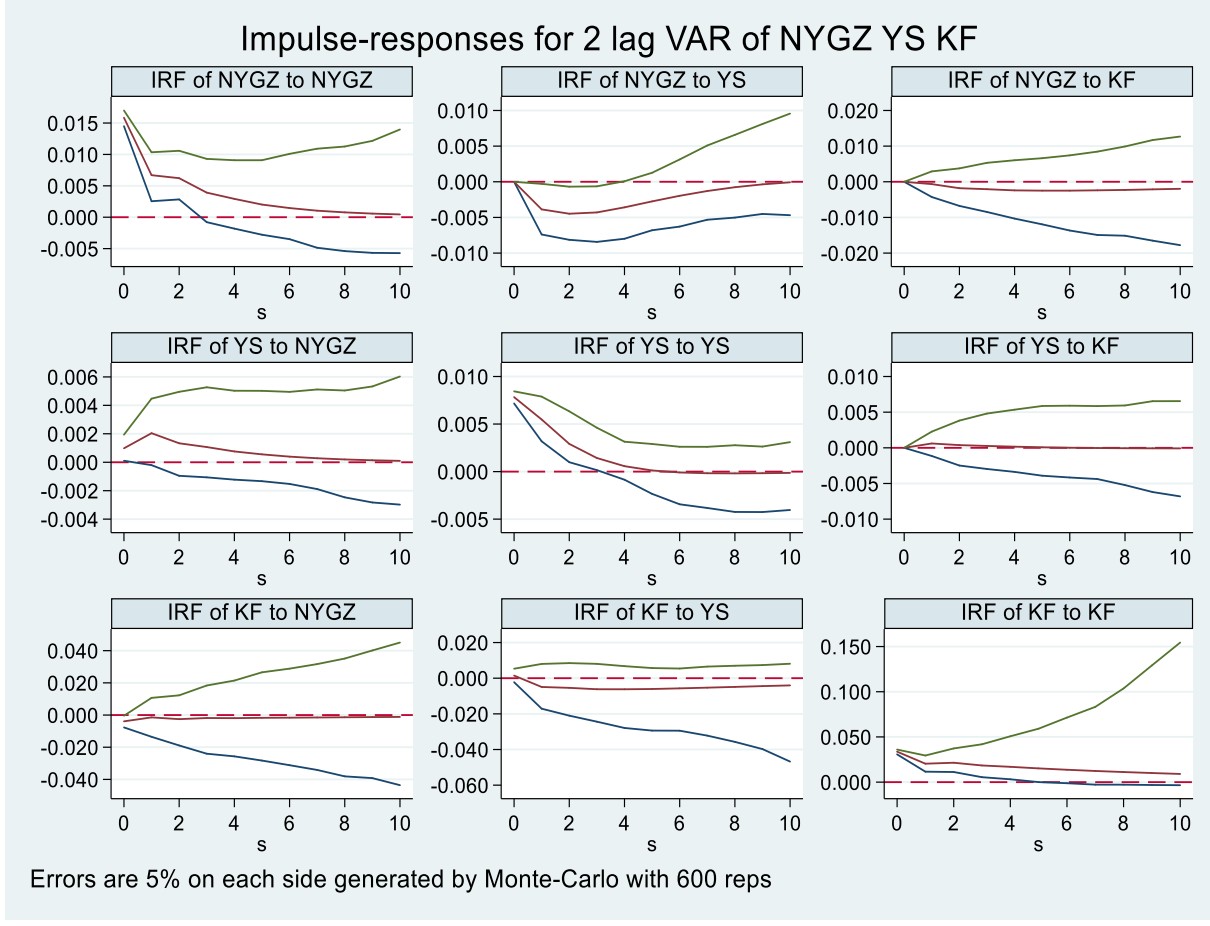

**Figure 3.** Impulse response diagram of NYGZ, YS and KF.

1.  For the response of the business environment (YS) and agricultural opening-up (KF) to the impulse shock from the agricultural high-quality development (NYGZ), the first

picture in the first row reflects that NYGZ showed a positive response but continued to decline under its own pulse shock, and gradually converged to 0 in the 10th period. The second graph in the first row reflects that, under a pulse shock of NYGZ, YS showed a negative response at the beginning and continued to strengthen, reached a negative peak in the second period and then gradually increased and converged to 0 in the tenth period. The third picture in the first row reflects that KF showed a weak negative response under a pulse shock of NYGZ. Although it was basically close to 0 in the 10th period, it did not tend to 0. It shows that a change in agricultural opening-up (KF) has a long-term impact on high-quality agricultural development (NYGZ).

2.  For the response of high-quality development of agricultural economy (NYGZ) and agricultural opening-up (KF) to the impulse shock from the business environment (YS), the second graph in the first column reflects that, under the impulse of YS, NYGZ gradually reached the highest positive response in the first period, but then it continued to decline and gradually converged to 0 in the ninth period. The second graph in the second column reflects that, under the impact of one pulse of YS itself, the positive response reached the highest level in the first period, but then it continued to decline and gradually converged to 0 in the fifth period. The second picture in the third column reflects that KF showed a weak positive response at the beginning under a pulse of YS, and then it gradually converged to 0 in the fourth period.

3.  For the response of high-quality agricultural economic development (NYGZ) and business environment (YS) to the impulse shock from agricultural opening-up (KF), the third picture in the first column reflects the weak response of NYGZ under a KF pulse shock, which basically tends to 0 in the fourth period. The third picture in the second column reflects that KF showed a small negative response under a pulse shock of YS. In the 10th period, although the small negative response was close to 0, it did not tend to 0. It shows that the negative impact of openness (KF) on the business environment (YS) is long-lasting. The third picture in the third column reflects the maximum positive response of KF at the beginning under the impact of one pulse of itself. In the 10th period, although the smaller positive response is close to 0, it does not tend to 0. It shows that the positive impact of an agricultural opening-up (KF) on its own changes is long-lasting.

AH2, BE2 and AO2 are the data of NYGZ, YS and KF after second-order difference, and then first-order lag processing, as shown in Figure 4.

After removing the individual characteristics of agricultural high-quality development (AH2), it first decreased and then increased after a pulse from itself, showing fluctuations up and down, but the impact continued to weaken and shrink, and it gradually converged and tended to 0 in the 10th period. After being impacted by the business environment (BE2), it first increased and then decreased, and after being impacted by the agricultural opening-up (AO2), it first decreased and then increased at 0.

After removing the individual characteristics of the business environment (BE2), under the impact of one of its own pulses, it gradually reached the highest positive response in the first period, but then the effect continued to decrease in waves, and began to gradually decrease in the sixth period. Under the impact of a pulse from AH2 and AO2, BE2 gradually increased first, but then it continued to decrease, and then the effect continued to decrease in waves and gradually converged to 0 in the tenth period.

After the agricultural opening-up to remove the individual characteristics (AO2), under the impact of one of its own pulses, it gradually reached the highest positive response in the first period, but then the effect continued to decrease in waves and began to gradually decrease in the sixth period. Under the impact of a pulse from AH2, BE2 gradually increased first, and then it decreased continuously and gradually converged to 0 in the seventh period.

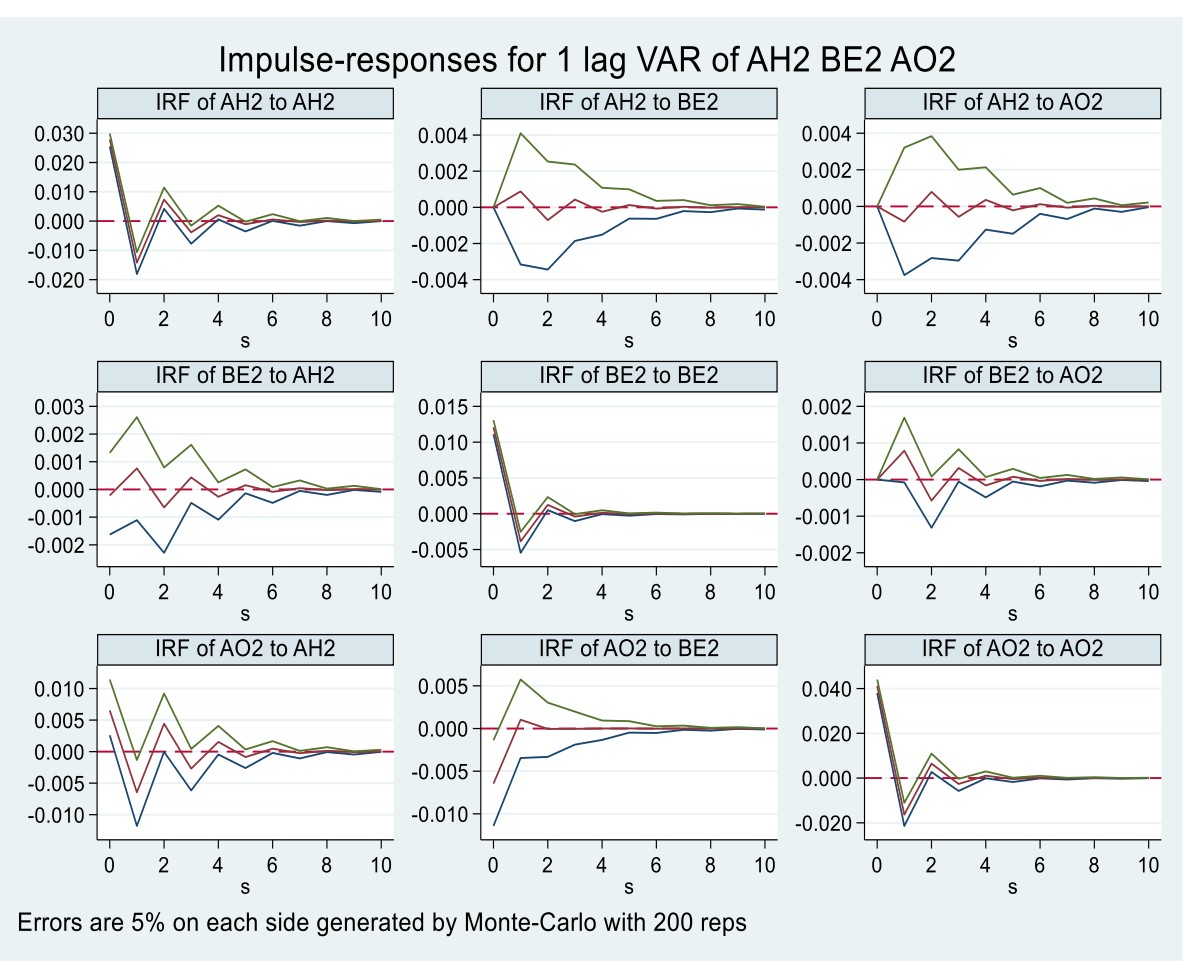

**Figure 4.** Impulse response diagram of AH2, BE2 and AO2.

*4.5. Analysis of Variance Decomposition*

In the PVAR model, the variance decomposition can explain the contribution of each structural shock to the endogenous variables, and this is complementary to the impulse response analysis. In order to more accurately analyze the degree of mutual influence and timeliness between high-quality agricultural development (NYGZ), business environment (YS) and agricultural opening-up (KF), the variance decomposition method is used to further measure the impact of each variable on the "self" and the proportion of changes caused by the impact of other variables, that is, the relative variance contribution rate. In order to more accurately reflect the dynamic changes of the relative variance contribution rate, Table 9 presents the variance decomposition results for the first to tenth forecast period. The first line is the response variable (the response variable is the shock variable). The IV (impulse variable) in the second line is the abbreviation of the pulse variable, and s is the number of lag periods; see Table 9 for details.

As can be seen from Table 9 as a whole, the results of the seventh to tenth periods as the impacted variables of NYGZ and KF are not very different. It shows that, after the seventh period, the forecast variance decomposition system is relatively stable, and continuing to extend the forecast period has little effect on the results.

This paper analyzes the changes in the business environment (YS) and agricultural opening-up (KF) caused by changes in the high-quality development of agricultural economy (NYGZ). From the first period, the change of NYGZ was completely affected by itself, not affected by YS and KF. After that, its own influence quickly decreased, and after the sixth period, it basically stabilized within 0.10, and the impact changed very much. small. The change of YS caused the change of NYGZ to increase at first, and then it reached the



highest level in the third period and then gradually decreased. The tenth period dropped below 0.001, and its impact became minimal. The change of NYGZ caused by the change of KF was relatively small at the beginning, and then it increased rapidly and rose to above 0.90 in the seventh period, and basically stabilized at about 0.91 in the following periods.

**Table 9.** Variance decomposition results of NYGZ, YS and KF.

| s IV | Response Variable NYGZ | | | Response Variable YS | | | Response Variable KF | | |
|---|---|---|---|---|---|---|---|---|---|
| | NYGZ | YS | KF | NYGZ | YS | KF | NYGZ | YS | KF |
| 1 | 1 | 0 | 0 | 0.0000104 | 0.9999897 | 0 | 0.0342366 | 0.0165645 | 0.9491989 |
| 2 | 0.7490986 | 0.1852139 | 0.0656874 | 0.1208248 | 0.8405502 | 0.0386250 | 0.0322154 | 0.0101391 | 0.9576455 |
| 3 | 0.4695218 | 0.2453484 | 0.2851298 | 0.2011365 | 0.7079589 | 0.0909046 | 0.0663013 | 0.0041822 | 0.9295165 |
| 4 | 0.2575728 | 0.1759909 | 0.5664363 | 0.2156136 | 0.6484010 | 0.1359855 | 0.0783882 | 0.0018488 | 0.9197630 |
| 5 | 0.1360786 | 0.0848674 | 0.7790541 | 0.1880390 | 0.6460492 | 0.1659119 | 0.0880078 | 0.0007237 | 0.9112684 |
| 6 | 0.0886728 | 0.0356071 | 0.8757201 | 0.1791642 | 0.6204193 | 0.2004165 | 0.0892570 | 0.0004401 | 0.9103029 |
| 7 | 0.0811575 | 0.0143779 | 0.9044647 | 0.1958859 | 0.5333037 | 0.2708104 | 0.0888738 | 0.0004055 | 0.9107208 |
| 8 | 0.0847818 | 0.0056201 | 0.9095981 | 0.2017466 | 0.3883259 | 0.4099274 | 0.0878267 | 0.0003838 | 0.9117895 |
| 9 | 0.0880974 | 0.0021886 | 0.9097140 | 0.1722002 | 0.2287895 | 0.5990102 | 0.0872253 | 0.0003520 | 0.9124227 |
| 10 | 0.0889501 | 0.0009799 | 0.9100700 | 0.1300360 | 0.1109987 | 0.7589653 | 0.0869598 | 0.0003229 | 0.9127173 |

The changes in the business environment (YS) caused by changes in the high-quality development of agricultural economy (NYGZ) and agricultural opening up (KF) are discussed in this paper. From the first period, the change of YS was almost entirely affected by itself, and then it gradually decreased slowly. In the tenth period, it was reduced to about 0.10, and the impact has become relatively small. The change of NYGZ caused the change of YS to be high and low, undulating in waves, but its volatility gradually shrank. The YS change caused by the KF change was 0 at the beginning, and then it gradually increased, and the 10th period rose to above 0.75.

This paper analyzes the changes in agricultural opening-up (KF) caused by changes in the high-quality development of agricultural economy (NYGZ) and business environment (YS). From the first period, the change of KF was 95% affected by itself, and then it gradually decreased. In the fifth period, it was reduced to about 0.91, and the subsequent periods have basically stabilized at around 0.91. The change in KF caused by the change in NYGZ is relatively small, but its trend first rises and then falls. The change in KF caused by the change of YS was very small, and then it gradually decreased. After the eighth period, it was basically stable at the level of 0.0003, and its impact become minimal.

After analyzing the results of high-quality agricultural economic development (AH2), business environment (BE2) and agricultural opening to the outside world (AO2) by the second-order difference, the variance decomposition method is used to further measure the change ratio of each variable when it is impacted by "itself" and other variables, that is, the relative variance contribution rate. It reflects the dynamic change of the relative variance contribution rate. Table 10 presents the variance decomposition results for the first to tenth forecast periods. In strong contrast with Table 9, from the first period to the tenth period, the change of each variable is almost completely affected by itself, and then gradually decreases; even in the tenth period, the decrease is relatively small, and the impact is relatively small. It is still very large. At the same time, it is affected by other variables. Although most cases increase from period to period, the proportion is very small and can almost be ignored.

**Table 10.** Variance decomposition results of AH2, BE2 and AO2.

| s IV | Response Variable AH2 | | | Response Variable BE2 | | | Response Variable AO2 | | |
|------|------|------|------|------|------|------|------|------|------|
| | AH2 | BE2 | AO2 | AH2 | BE2 | AO2 | AH2 | BE2 | AO2 |
| 1 | 1 | 0 | 0 | 0.0073245 | 0.9926755 | 0 | 0.0118632 | 0.0058936 | 0.9822433 |
| 2 | 0.9993217 | 0.0006758 | 0.0000026 | 0.0191789 | 0.9772435 | 0.0035776 | 0.01533201 | 0.0051005 | 0.9795794 |
| 3 | 0.9988187 | 0.0011685 | 0.0000128 | 0.0276983 | 0.9667686 | 0.0055330 | 0.0177110 | 0.0049190 | 0.9773700 |
| 4 | 0.9985353 | 0.0014396 | 0.0000250 | 0.0325333 | 0.9612917 | 0.0061750 | 0.0191358 | 0.0048834 | 0.9759808 |
| 5 | 0.9983869 | 0.0015784 | 0.0000347 | 0.0350582 | 0.9585960 | 0.0063475 | 0.0199178 | 0.0048778 | 0.9752045 |
| 6 | 0.9983109 | 0.0016479 | 0.0000412 | 0.0363377 | 0.9572767 | 0.0063857 | 0.0203288 | 0.0048773 | 0.9747939 |
| 7 | 0.9982725 | 0.0016825 | 0.0000450 | 0.0369789 | 0.9566273 | 0.0063937 | 0.0205401 | 0.0048775 | 0.9745824 |
| 8 | 0.9982531 | 0.0016997 | 0.0000472 | 0.0372991 | 0.9563060 | 0.0063937 | 0.0206473 | 0.0048777 | 0.9744750 |
| 9 | 0.9982434 | 0.0017083 | 0.0000483 | 0.0374588 | 0.9561466 | 0.0063947 | 0.0207014 | 0.0048778 | 0.9744208 |
| 10 | 0.9982385 | 0.0017126 | 0.0000489 | 0.0375384 | 0.9569671 | 0.0063944 | 0.0207286 | 0.0048778 | 0.9743936 |

It can be seen from the above variance decomposition results that the actual data values are used to analyze the variance ratio of the prediction error, and the importance of different structural shocks can be further evaluated by analyzing the endogenous changes of each structural shock (measured by variance). It further evaluates the relative magnitude of the importance of different structural shocks. The variance decomposition above gives information on the relative importance of each random perturbation that affects the response variables in the PVAR model for high-quality agricultural development (NYGZ), business environment (YS) and agricultural opening-up (KF). As time continues to predict, those influences are getting stronger and stronger.

It is clear from Table 9 that China's agricultural opening to the outside world plays a crucial role in the long-term development of both the high-quality development of agricultural economy and the business environment, as well as the opening to the outside world itself. Therefore, we should give priority to the development of agricultural opening to the outside world and build a new development pattern of agricultural opening to the outside world. At the same time, we will promote high-quality development of agriculture and improve the business environment. With a view to serving China's overall diplomacy and agricultural and rural development, we will speed up the building of a new type of agricultural cooperative partnership with foreign countries and promote the Belt and Road Initiative with high quality. Through agricultural rural cooperation, we can optimize the layout of agricultural trade; expand agricultural foreign investment; deepen China's agricultural science and technology cooperation; actively promote the depth of China in the global food and agriculture management; accurately implement agricultural foreign aid; achieve higher levels of agricultural opening to the outside world, promoting rural revitalization; and make a positive contribution to speed up the agricultural modernization.

In the post-epidemic era, the Chinese government should give priority to deepening cooperation in agricultural science and technology in implementing its agricultural opening-up policy, and, at the same time, enhance its capacity for international collaborative innovation. When focusing on the weak spots in agricultural science and technology, we should continue to focus on the major areas. In terms of international joint research on basic research, it promotes significant theoretical breakthroughs in biological breeding, excellent seed selection, plant protection, animal husbandry and veterinary medicine, resources and environment, agricultural machinery and equipment, etc. In the field of frontier interdisciplinary research, China will carry out interdisciplinary research in the fields of synthetic biology, remote sensing, big data, artificial intelligence and rural human settlement environment governance. In terms of key agricultural core technologies, China will promote joint research and development related to the needs of industrial development, especially domestic and international joint research and development on carbon-peak and carbon-neutral technologies.

In particular, China should strengthen the construction of international cooperation platforms to provide effective carriers for international agricultural science and technology cooperation. China will establish experimental and demonstration bases for agricultural technologies at home and abroad; give priority to scientific and technological cooperation in the Belt and Road countries in the prevention and control of animal and plant diseases, high-efficiency crop planting and the formulation of agricultural quality standards; and form overseas integrated demonstration of agricultural technologies to provide scientific and technological support for agricultural enterprises going global. China encourages agricultural science and technology to "go global". As a bridge, the alliance promotes the cross-border flow of agricultural science and technology innovation elements and improves the global cooperation network system of agricultural science and technology in China.

## 5. Conclusions and Suggestions

The above analysis shows that the high quality of China's agricultural economy, the business environment and the opening up of agriculture are mutually reinforcing and indispensable. Therefore, China should strengthen and develop these three factors. It should not only focus on the individual indicators at that time, but also focus on the long-term effects and long-term benefits of investment. At the same time, it will strengthen and optimize other economic environments and enhance the upgrading and driving of domestic consumption. The NYGZ, YS and KF impulse response maps from the national data are more obvious than the NYGZ, YS and KF impulse response maps from the data of the five northwestern provinces. YGZ, YS and KF are not only affected by the local area, but also mainly by the national environment. Therefore, in the construction of high-quality agricultural economy, business environment and agricultural opening in the five northwestern provinces and regions, attention should be paid to changes in other provinces and municipalities. In addition to the impact on the northwest region, it is necessary to pay more attention to the close cooperation within the five northwest provinces and regions. Therefore, it should be developed in a coordinated manner, and no one aspect of development should be neglected. At the same time, it also gives us obvious inspiration to strengthen the cooperation between the five northwestern provinces, as this will help save a lot of logistics and transportation costs and form a scale effect and superposition effect.

In short, the high-quality development of modern agriculture is a comprehensive dynamic system which involves many factors and elements. It not only includes time and space effects, but also considers various factors, conditions, induced effects between elements, and the local characteristics and endowments of each region. At this stage, in order to rapidly promote the high-quality development of the agricultural economy, first of all, it is necessary to ensure its external environment and internal driving force. To optimize the external environment for the high-quality development of the agricultural economy is to optimize the business environment first. Today, China's economy has entered the context of a new development pattern that takes the domestic cycle as the main body, and the domestic and international dual cycles promote each other. Therefore, the high-quality development of China's agricultural economy must be driven by the agricultural opening-up. Agricultural opening-up not only has a strong long-term driving force for itself, but also continues to strengthen its driving force for the high-quality development of the agricultural economy and the business environment as time goes on, and the effect is becoming more and more obvious.

The limitation of this paper is that it only considers the impact of regional business environment and agricultural opening-up on high-quality agricultural development, ignoring the impact of other control variables. This may lead to certain biases in the results. In addition, the analysis of this paper is not divided into regions. During the writing process, it was found that the level of agricultural development in different regions is different, and the factors affecting agricultural development are also different. Therefore, the following research can analyze the provinces by agricultural regions. Finally, a comparative analysis of the business environment and agricultural opening-up in China and foreign countries

can be conducted to gain experience and inspiration for the high-quality development of China's agriculture.

**Author Contributions:** Conceptualization, D.W.; methodology, D.W.; writing—original draft preparation, D.W.; writing—review and editing, D.W., B.A., Q.L., Y.L. and Y.Z. All authors have read and agreed to the published version of the manuscript.

**Funding:** This research was funded by the National Social Science Foundation of China (No. 19XGL007 and No. 19BJY139), the Humanities and Social Sciences Fund of the Ministry of Edu-cation of China (No. 17YJAZH057).

**Institutional Review Board Statement:** Not applicable.

**Informed Consent Statement:** Not applicable.

**Data Availability Statement:** Not applicable.

**Conflicts of Interest:** The authors declare that there is no conflict of interest.

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
