# Peer review of "Regional Business Environment, Agricultural Opening-Up and High-Quality Development: Dynamic Empirical Analysis from China’s Agriculture"

_agronomy, doi:10.3390/agronomy12040974_

Round 1
Reviewer 1 Report
The manuscript analyzes the interaction between business environment, agricultural opening-up and high-quality agricultural economy from the perspective of China's provinces. The introduction provide sufficient background and include relevant references.
However there are some problems raised:
1) tables 1,2,3 are not clearly presented and there is a language error in table 1,
2) Further research directions should be given and discussion,
3) The discussion on the existing work is not sufficient.
Reviewer 2 Report
The article is good and almost ready for publication in Agronomy. Just make these two inclusions of references.
1. What is the corona epidemic? Is it covid-19? If so, I would like these 2 articles to be cited here in line 53:
-Effects of COVID-19 on Chinese sectoral indices: a multifractal analysis
FHA De AraÚJo, LHS Fernandes, BM Tabak
Fractals 29 (7), 2150198-587
-Insights from the (in) efficiency of Chinese sectoral indices during COVID-19
LHS Fernandes, FHA de Araujo, BM Tabak
Physica A: Statistical Mechanics and its Applications 578, 126063
Round 2
Reviewer 1 Report
The revisions are adequate. The tables are clearly presented, language errors were corrected and further research directions and discussion were added.